# Effect of α-Al_2_O_3_ Additive on the Surface Micro-Arc Oxidation Coating of Ti6Al4V Alloy

**DOI:** 10.3390/nano13111802

**Published:** 2023-06-05

**Authors:** Yuke Chen, Meini Yuan

**Affiliations:** College of Mechatronic Engineering, North University of China, Taiyuan 030051, China; baiye1823936023@163.com

**Keywords:** Ti6Al4V alloy, micro-arc oxidation, α-Al_2_O_3_ particles, frictional wear properties, corrosion resistance

## Abstract

α-Al_2_O_3_ nanoparticles can enter a micro-arc oxidation coating and participate in the coating-formation process through chemical reaction or physical–mechanical combination in the electrolyte. The prepared coating has high strength, good toughness and excellent wear and corrosion resistance. In this paper, 0, 1, 3 and 5 g/L of α-Al_2_O_3_ nanoparticles were added to a Na_2_SiO_3_-Na(PO_4_)_6_ electrolyte to study the effect on the microstructure and properties of a Ti6Al4V alloy micro-arc oxidation coating. The thickness, microscopic morphology, phase composition, roughness, microhardness, friction and wear properties and corrosion resistance were characterized using a thickness meter, scanning electron microscope, X-ray diffractometer, laser confocal microscope, microhardness tester and electrochemical workstation. The results show that surface quality, thickness, microhardness, friction and wear properties and corrosion resistance of the Ti6Al4V alloy micro-arc oxidation coating were improved by adding α-Al_2_O_3_ nanoparticles to the electrolyte. The nanoparticles enter the coatings by physical embedding and chemical reaction. The coatings’ phase composition mainly includes Rutile-TiO_2_, Anatase-TiO_2_, α-Al_2_O_3_, Al_2_TiO_5_ and amorphous phase SiO_2_. Due to the filling effect of α-Al_2_O_3_, the thickness and hardness of the micro-arc oxidation coating increase, and the surface micropore aperture size decreases. The roughness decreases with the increase of α-Al_2_O_3_ additive concentration, while the friction wear performance and corrosion resistance are improved.

## 1. Introduction

Titanium alloys have a wide range of applications in aerospace, shipbuilding and biomedical fields because of their light mass, high specific strength and good thermal stability [1,2,3]. However, titanium alloy has low hardness. When applied as an aerospace material, it is prone to scratching and has a large bite tendency, and is easily damaged by the scraping of abrasive particles during sliding friction, and abrasive wear occurs [4]. In addition, the surface integrity of a Ti6Al4V alloy is sensitive, and surface damage is very likely to induce the sprouting and expansion of fatigue cracks, which greatly limits the alloys’ application under severe friction and other working conditions [5,6]. In addition, the standard potential of a titanium electrode is only −1.63 V (relative to a standard hydrogen electrode—NHE), so it is susceptible to various forms of corrosion such as galvanic corrosion, crevice corrosion, intergranular corrosion and stress corrosion cracking under the action of chemical or electrochemical reactions [7,8]. Surface-strengthening processes can greatly improve the surface hardness of titanium alloys without changing the internal structure and mechanical properties of the material. Commonly used surface-strengthening processes for titanium alloys include thermal oxidation, laser melting, ion implantation, anodic oxidation and micro-arc oxidation (MAO) [9,10,11,12]. Among them, micro-arc oxidation technology has attracted a lot of attention from scholars all over the world because of its easy operation, green credentials and excellent coating performance [13,14,15,16]. By choosing the electrolyte solution and adjusting the electrical parameters of the micro-arc oxidation process, a highly dense and well-bonded ceramic-phase oxidation coating can be grown on the surface of Ti, Al, Mg and their alloy substrates, which can significantly improve the surface hardness and wear and corrosion resistance [17,18,19,20,21].

As the micro-arc oxidation process grows a large number of fine holes on the surface of the substrate material, it will inevitably reduce the protection performance of the micro-arc oxidation coating, which will affect the service performance of the material in some special environments [22,23,24,25]. Recent studies on the addition of nano-additives to a micro-arc oxidation electrolyte solution have shown that α-Al_2_O_3_ nanoparticles are easily dispersed in the electrolyte, especially in alkaline solutions, and can participate in the coating-formation process by chemical reaction or physical–mechanical combination [26,27,28,29]. The prepared coating has high strength, good toughness and excellent wear and corrosion resistance. Niazi [1] prepared a TiO_2_-Al_2_O_3_ nanocomposite coating on the surface of pure titanium by an electrophoresis-enhanced micro-arc oxidation technique, which resulted in a fourfold increase in surface hardness and a significant increase in corrosion resistance. Shokouhfar [30] investigated the effect of α-Al_2_O_3_ nanoparticles on the micro-arc oxidation coating of pure titanium. The results showed that the friction coefficient decreases to 0.5, and the corrosion current density decreases from 1.7 × 10^−6^ A/cm^2^ to 1 × 10^−6^ A/cm^2^.

In this paper, the electrolyte solution was modified by adding α-Al_2_O_3_ nanoparticles with different mass concentrations (0, 1, 3 and 5 g/L) to the electrolyte, and the effects of the concentration variation on the thickness, microscopic morphology, phase composition, roughness, microhardness, friction and wear properties and corrosion resistance of the Ti6Al4V alloy micro-arc oxidation coatings were investigated to explore the mechanism of friction reduction and anti-corrosion, so as to lay the foundation for further expanding the application of Ti6Al4V alloys.

## 2. Experimental Materials and Methods

### 2.1. Micro-Arc Oxidation Coating Preparation

The experiment material was a Ti6Al4V alloy with a chemical composition of 6.3 wt% Al, 4.2 wt% V, 0.15 wt% O, 0.11 wt% Fe, 0.03 wt% C, 0.02 wt% N, 0.001 wt% H and balance Ti. Specimens with a size of 50 mm × 30 mm × 6 mm were ground using 200^#^, 400^#^, 800^#^ and 1200^#^ SiC water sandpaper. The surface oil was removed by ultrasonic cleaning with anhydrous ethanol for 15 min, then rinsing with deionized water and drying.

A model of a micro-arc oxidation power supply was used, with the electrolytic cell as the cathode and the specimen as the anode. The power supply was operated by selecting a constant voltage mode of 500 V for 20 min. The pulse frequency and duty ratio were 600 Hz and 20%, respectively. The electrolyte was prepared using Na_2_SiO_3_ (8 g/L), Na(PO_4_)_6_ (3 g/L), NaOH (1 g/L) and C_3_H_8_O_3_ (6 mL/L) in distilled water. α-Al_2_O_3_ nanoparticles (0, 1, 3 or 5 g/L) with 150 nm particle size were added.

### 2.2. Organizational Structure Analysis and Performance Testing

A scanning electron microscope (SEM, MIRA4, TESCAN, Brno, Czech Republic) was used to observe the microscopic morphology of the micro-arc oxidation coatings. An X-ray diffractometer (XRD, Empyrean, Panaco, Almelo, The Netherlands) was used to analyze the phase structure. The thickness was measured using a coating thickness tester (ATS230, Test TIME, Beijing, China). The surface roughness was characterized using a confocal microscope (LSM900, ZEISS, Oberkochen, Germany). A microhardness tester (HVS-1000B, Zhongte, Dongguan, China) was used with a load of 500 g and 10 s. The tribological properties of the samples were tested using a UMT-2 tribological wear tester (UMT-2, CETR, San Jose, CA, USA) with a load of 2 N. The Tafel polarization curves of the Ti6Al4V alloy substrate and its micro-arc oxide film were tested using a CHI600E electrochemical workstation to obtain the corrosion current density, *i*_corr_, and corrosion potential, *E*_corr_, and to calculate the polarization resistance, *R*_p_, of the film.

## 3. Results and Discussion

### 3.1. Effect of α-Al_2_O_3_ Concentration on the Thickness of Micro-Arc Oxidation Coatings

Figure 1 shows the variation of thickness of micro-arc oxidation coatings on Ti6Al4V alloys with different mass concentrations of α-Al_2_O_3_. The thickness increases slightly with the increase of α-Al_2_O_3_ concentration, and the maximum thickness is 50.22 μm. This is due to the increased conductivity of the electrolyte as a result of the addition of α-Al_2_O_3_ nanoparticles, which increases the ion concentration in the micro-arc oxidation process. The Ti6Al4V alloy produces more arc discharge centers on the surface, making the micro-arc oxidation process more intense. On the other hand, during the liquid-quenching process, α-Al_2_O_3_ nanoparticles can be used as nucleation cores to accelerate the coating growth, and participate in the coating-formation reaction through filling.

### 3.2. Effect of α-Al_2_O_3_ on the Microscopic Morphology of Micro-Arc Oxidation Coating

Figure 2 and Figure 3 show the microscopic morphology and surface roughness variation of Ti6Al4V alloy micro-arc oxidation coatings under different α-Al_2_O_3_ nanoparticle concentrations. When the α-Al_2_O_3_ nanoparticle concentration is 3 g/L (as in Figure 2c), the number and size of micropores on the surface decreases, and the defects, such as microcracks, are significantly reduced. It can be inferred that the addition of an appropriate amount of α-Al_2_O_3_ nanoparticles to the electrolyte helps to improve the coating densities and defects. This is because the α-Al_2_O_3_ nanoparticles increase the electrical conductivity of the electrolyte solution, and their high surface activity. A large number of ions are attracted by the nanoparticles and agglomerate on the surface of the Ti6Al4V alloy during the reaction process, which promotes the micro-arc oxidation reaction, and increases the micro-arc discharge energy and the size of the individual arc discharge areas. The molten material spreads uniformly on the surface of the substrate. As such, the number of pores is reduced, which is consistent with the more intense reaction and increased intensity of arc light during the micro-arc oxidation process. Meanwhile, some α-Al_2_O_3_ nanoparticles are adsorbed onto the surface and embed into the coating during the reaction, resulting in the channels generated by the arc discharge being repaired by the particle volume effect.

### 3.3. Effect of α-Al_2_O_3_ Concentration on the Physical Phase Composition of Micro-Arc Oxidation Coatings

Figure 4 shows the XRD patterns of Ti6Al4V alloy micro-arc oxidation coatings with different mass concentrations of added α-Al_2_O_3_ nanoparticles. The diffraction peak intensity of the original phase slightly changes with the increase of the addition of α-Al_2_O_3_ nanoparticles. Compared with the coating prepared in the electrolyte without α-Al_2_O_3_, the intensity of the Rutile-TiO_2_ diffraction peaks in the coating prepared with α-Al_2_O_3_ was increased. This is due to the increase in electrolyte conductivity because of the aggregation of α-Al_2_O_3_ nanoparticles. This results in an increase in arc-discharge energy and a higher instantaneous sintering reaction temperature in the microzone, which is favorable for the high-temperature conversion of Anatase-TiO_2_. In addition, α-Al_2_O_3_ and Al_2_TiO_5_ phases also appear in the coating prepared by adding α-Al_2_O_3_ nanoparticles to the electrolyte.

Diffraction peaks of titanium metal were observed in the XRD pattern from the matrix Ti6Al4V alloy, which is due to X-ray penetration through the micropores and discharge channel to the matrix. When the thickness of the coating is deeper, the X-ray penetration becomes difficult and the intensity of the titanium diffraction peaks is reduced. The XRD pattern contains a typical amorphous material “bun peak” form in the interval of 15°–25°. The silicon oxide and phosphide diffraction peaks are not found. It is presumed that some of the oxides containing SiO_2_ and phosphorus exist in the coating in the form of an amorphous phase.

### 3.4. Effect of α-Al_2_O_3_ Concentration on the Microhardness of Micro-Arc Oxidation Coatings

The effects of α-Al_2_O_3_ additives on the microhardness of Ti6Al4V alloy micro-arc oxidation coatings at different concentrations are shown in Figure 5. With the increase of α-Al_2_O_3_ concentration, the microhardness of the coating showed a varied pattern of first increasing and then slightly decreasing. When the concentration was 3 g/L, the microhardness reached a maximum of 903.58 HV, which is a 161% increase compared with the Ti6Al4V alloy substrate. Combined with the analysis of the thickness, surface morphology and phase composition of the micro-arc oxidized composite, it can be seen that the addition of α-Al_2_O_3_ nanoparticles makes the coating structure more dense and generates more of a high-hardness Rutile-TiO_2_ phase, which increases the microhardness of the film layer to a certain extent. However, its impact is limited and the increase of the microhardness is small. When the concentration of α-Al_2_O_3_ nanoparticles was 5 g/L, the surface structure produced defects, which led to a decrease of the composite film layer microhardness.

### 3.5. Effect of α-Al_2_O_3_ Concentration on the Frictional Wear Properties of Micro-Arc Oxidation Coatings

Figure 6 shows the changes of friction factors of the prepared micro-arc oxidation coatings and Ti6Al4V alloy under dry friction sliding conditions at different α-Al_2_O_3_ nanoparticle concentrations. It can be seen that the friction factor of the Ti6Al4V alloy matrix starts to increase from about 0.1 after the start of the friction test, and enters into a stable frictional wear stage after a rising period of approx. 80 s, with a friction factor of approx. 0.61.

Figure 7 is a schematic diagram of the friction mechanism of a micro-arc oxidation coating. Due to the sparse surface layer roughness, with a more uneven microporous structure and protruding ceramic particles, contact of a GCr15 steel ball with the micro-arc oxidation coating generated an impact vibration. The stress is first concentrated in the outermost sparse raised layer, making the brittle ceramic materials sensitive to the repeated impact effect produced by crushing. Serious damage to the outer oxide layer occurs and sharp ceramic abrasive chips are produced. Some of the abrasive chips that accumulate during the subsequent repeated grinding and pushing process in the sliding surface, accumulate into the low-lying pits, and GCr15 steel ball grinding wear occurs. The extremely high hardness of the ceramic phase makes part of the material adhere to the anti-wear interface after plowing of the steel ball, and the wear mechanism changes to abrasive wear and adhesive wear caused by material transfer. As a result, the friction curve of a micro-arc oxidation coating specimen without the addition of α-Al_2_O_3_ nanoparticles jittered more strongly, and the friction factor in the stable period increased to 0.72, which is higher than that of a Ti6Al4V alloy without surface-strengthening treatment. It shows that the micro-arc oxidation coating has a good protective effect on the substrate.

The friction factor of the micro-arc oxidation coating prepared at a concentration of 3 g/L of α-Al_2_O_3_ nanoparticles increased rapidly to a steady state after the start of the friction test, with a friction factor of 0.41, which was 0.2 lower compared to that of the Ti6Al4V alloy. Figure 8 shows the friction mechanism of the micro-arc oxidation coating containing α-Al_2_O_3_ nanoparticles and a GCr15 steel ball, showing resistance against friction. The wear resistance is related to the alloy’s structure and hardness, which generates more hard Rutile-TiO_2_ and introduces new Al_2_TiO_5_. They provide hard bearings for the coating, and avoids its excessive wear loss. Meanwhile, the α-Al_2_O_3_ nanoparticles are evenly dispersed inside the coating, and during frictional contact with the GCr15 steel ball, the α-Al_2_O_3_ nanoparticles undergo shear and transfer, and reduce the friction of the sliding interface during the friction process. The friction property of some areas is changed from sliding friction to rolling friction, which reduces the friction factor.

### 3.6. Effect of α-Al_2_O_3_ Concentration on the Corrosion Resistance of Micro-Arc Oxidation Coatings

The kinetic potential polarization curves of a Ti6Al4V alloy and micro-arc oxidation coatings prepared with different concentrations of α-Al_2_O_3_ additives are shown in Figure 9. Table 1 shows the fitted corrosion potential, *E*_corr_, corrosion current density, *i*_corr_, and linear polarization resistance, *R*_p_. Since the coating played a protective role for the titanium alloy by preventing the direct contact between a NaCl corrosive solution and the Ti6Al4V alloy, all the specimens treated by micro-arc oxidation showed a strong corrosion resistance compared with the Ti6Al4V alloy. The magnitude of the corrosion current density, *i*_corr_, of the coating prepared without α-Al_2_O_3_ is 6.63 × 10^−6^ A/cm^2^, which is an order of magnitude lower than the *i*_corr_ of the substrate (2.25 × 10^−5^ A/cm^2^). The corrosion potential has an increase of 226 mV.

Compared with a coating prepared without the addition of α-Al_2_O_3_ nanoparticles, the corrosion current density, *i*_corr_, of the coating was further reduced to 7.99 × 10^−7^ A/cm^2^ when the additive concentration was 3 g/L. Its linear polarization resistance, *R*_p_, was elevated to 5.85 × 10^4^ Ω/cm^2^, which was improved by one order of magnitude, and the corrosion potential was increased to 197 mV. The corrosion resistance was further enhanced, which can be attributed to the following:
The α-Al_2_O_3_ nanoparticles contained in the coating play a certain role in repairing the discharge channels (see Figure 10) and reducing the number of penetration holes. This makes the coating more dense, and effectively isolates the external corrosive environment from corrosive factors such as Cl^−^ ions, which plays an important role in improving the corrosion resistance.After the addition of α-Al_2_O_3_ nanoparticles, the coating contains new phases of α-Al_2_O_3_ and Al_2_TiO_5_ in addition to Anatase-TiO_2_ and Rutile-TiO_2_. The α-Al_2_O_3_ nanoparticles are diffusely distributed in the coating, while Al_2_TiO_5_ has a high hardness and strong corrosion resistance. Its isolation reduces the electrochemical corrosion tendency of the specimen and further improves the corrosion resistance.The thickness of micro-arc oxidation coatings prepared by adding an appropriate amount of α-Al_2_O_3_ to the electrolyte increases due to the increased electrolyte conductivity. It helps to slow the intrusion of corrosive media to the substrate to a certain extent.The growth process of the coating is mainly by the breakdown melting of the substrate and the transient cooling of the molten material in the electrolyte. Based on the good toughness of α-Al_2_O_3_ nanoparticles, embedding them in the coating is beneficial to reduce internal stress, which is helpful to improve the corrosion resistance.


### 3.7. Growth Mechanism of the Ti6Al4V Alloy Micro-Arc Oxide Coatings

The Ti6Al4V alloy micro-arc oxidation coating growth process is shown in Figure 11. At the initial stage of micro-arc oxidation, the applied voltage is lower than the arc starting voltage of the titanium alloy, and small bubbles are generated on the surface of the specimen with a piercing current sound, at which time the current flows to the cathode through the anode and the electrolyte reacts electrolytically under the action of the applied voltage, as in Equations (1) and (2).
4OH^−^ − 4e → 2H_2_O + O_2_↑(1)
2H_2_O → 2H_2_↑ + O_2_↑(2)

During the constant voltage mode, the oxidation voltage is rapidly increased to a set value, and under the action of a strong electric field, OH^−^ ions are ionized in the electrolyte, and Ti^4+^ is ionized in the titanium alloy substrate, both positive and negative ions are adsorbed onto the surface of the substrate and combine under the electric field force to form fused oxide TiO_2_ (as in Equations (3) and (4)), which is ejected outward through the discharge channel and uniformly distributed on the breakdown site of the specimen, and into the electrolyte. A ceramic porous coating is formed by solidification under the cooling effect of the electrolyte. The coating grows gradually by the cyclic discharge effect of the pulsed high-voltage electric field on the substrate.
Ti → Ti^4+^ + 4e (3)
Ti^4+^ + OH^−^ → TiO_2_ + 2H_2_O (4)

In the late stage of oxidation, the equivalent resistance of the micro-arc oxidation coating increases, electric breakdown occurs only at the microholes or weak positions on the surface, the high-brightness arc light appears locally and lasts longer, the growth rate of the coating decreases significantly and the internal phase change is dominated by a high temperature at this time.

From the above analysis, it can be seen that the electrolyte and the substrate material can not come into contact with each other due to blockage by insulating substances, such as passivation coating and gas barriers on the substrate surface. However, the discharge channel formed on the metal surface under the action of a strong electric field allows bilateral material reactions. While dielectric breakdown occurs, anions and cations migrate and react, generating new substances that cool and solidify on the alloy surface to form an initial oxide coating, which grows mainly inward as part of the substrate surface is melted at this time. The initial coating is formed in the weak microregion, and the process of “breakdown, reaction, condensation, and re-breakdown” continues, so the coating can continue to grow. The coating then thickens, and the growth direction also changes to outward growth.

When α-Al_2_O_3_ nanoparticles exist in the electrolyte, some α-Al_2_O_3_ nanoparticles are embedded in the coating by adsorption through electrophoretic force, molecular thermal motion and external impact. Meanwhile, excess α-Al_2_O_3_ nanoparticles and Ti^4+^ generated on the surface of the anode Ti6Al4V alloy, under the action of a strong electric field, undergo the chemical reaction of Equation (5) to generate Al_2_TiO_5_ phase.
TiO_2_ + α-Al_2_O_3_ (at an excess) → Al_2_TiO_5_·Al_2_O_3_ (1573 K) (5)

Figure 12 shows the growth curves of micro-arc oxidation coatings with different α-Al_2_O_3_ nanoparticle concentrations. The thickness of the micro-arc oxidation coating increases rapidly in the initial stage of oxidation with or without the addition of α-Al_2_O_3_ nanoparticles, and the slope of the coating thickness curve decreases and the growth rate decreases after about 500 s of oxidation time. Analysis shows that the growth rate of the micro-arc oxidation coating depends on the number of breakdown channels generated by the discharge. In the early stage of oxidation, dielectric breakdown occurs in a large number of micro-areas on the substrate surface, resulting in a large number of small discharge channels through which the electrolyte material and the molten material of the substrate metal can contact and react. However, the thickness of the film layer increases continuously with the oxidation process, and the number of discharge channels gradually decreases due to the increase of the equivalent resistance, *R*_p_, of the coating under a constant applied voltage, *E*, which is lower than the breakdown voltage, resulting in a decrease of the coating growth rate. The growth rate decreases continuously.

## 4. Conclusions

In this paper, α-Al_2_O_3_ nanoparticles were added to the electrolyte during micro-arc oxidation, and the effects of the mass concentration on the thickness, microscopic morphology, roughness, physical phase composition, microhardness, friction wear properties and corrosion resistance of the micro-arc oxidation coating were investigated. The conclusions are as follows:
The surface density of the coating is significantly improved by embedding α-Al_2_O_3_ nanoparticles, and the number of micropores and defects, such as grooves and pits, on the surface are reduced. The coating contains Ti, Anatase-TiO_2_ and Rutile-TiO_2_ phases. The diffraction peak intensity of the Rutile-TiO_2_ phase is improved by the addition of α-Al_2_O_3_ nanoparticles. In addition, some α-Al_2_O_3_ nanoparticles are embedded in the coating, and a new Al_2_TiO_5_ phase is generated.With the increase of α-Al_2_O_3_ nanoparticle concentration in the electrolyte, the thickness of the coating increased slightly, and reached a maximum value of 50.22 μm when the particle concentration was 5 g/L. The surface roughness of the coating decreased and then increased, and the roughness, *R*_a_, reached a minimum value of 3.82 μm when the particle concentration was 3 g/L; the surface was the flattest at this time. The microhardness of the coating increased, then decreased, and reached a maximum value of 903.58 HV when the concentration was 3 g/L.Compared with the Ti6Al4V alloy, the micro-arc oxidation coating has better wear resistance. The coating prepared with 3 g/L of α-Al_2_O_3_ nanoparticles has a stable friction factor of 0.41 and the best frictional wear performance, mainly because it has the lowest roughness and the highest surface hardness. The α-Al_2_O_3_ nanoparticles have a friction-reducing effect, and the surface micropores provide space for storage of generated abrasive chips.When the concentration of α-Al_2_O_3_ nanoparticles is 3 g/L, the good density and the isolation of Rutile-TiO_2_, α-Al_2_O_3_ and Al_2_TiO_5_ phases in the coating protect the substrate from corrosive solutions to a certain extent. The linear polarization resistance, *R*_p_, is increased to 5.85 × 10^4^ Ω/cm^2^, which is improved by one order of magnitude, and the corrosion potential of the specimen is 197 mV, an increase of 312 mV, and the corrosion current density, *i*_corr_, is 7.99 × 10^−7^ A/cm^2^, a decrease of two orders of magnitude.


## Figures and Tables

**Figure 1 nanomaterials-13-01802-f001:**
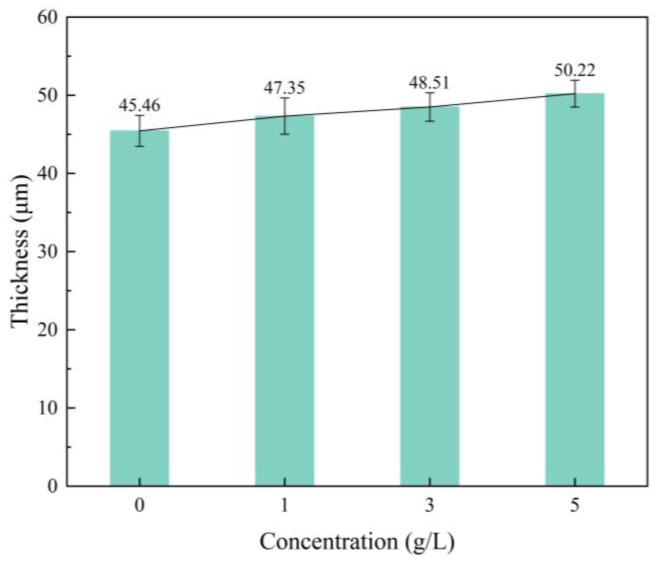
Thickness of micro-arc oxidation coating with different α-Al_2_O_3_ nanoparticle concentrations.

**Figure 2 nanomaterials-13-01802-f002:**
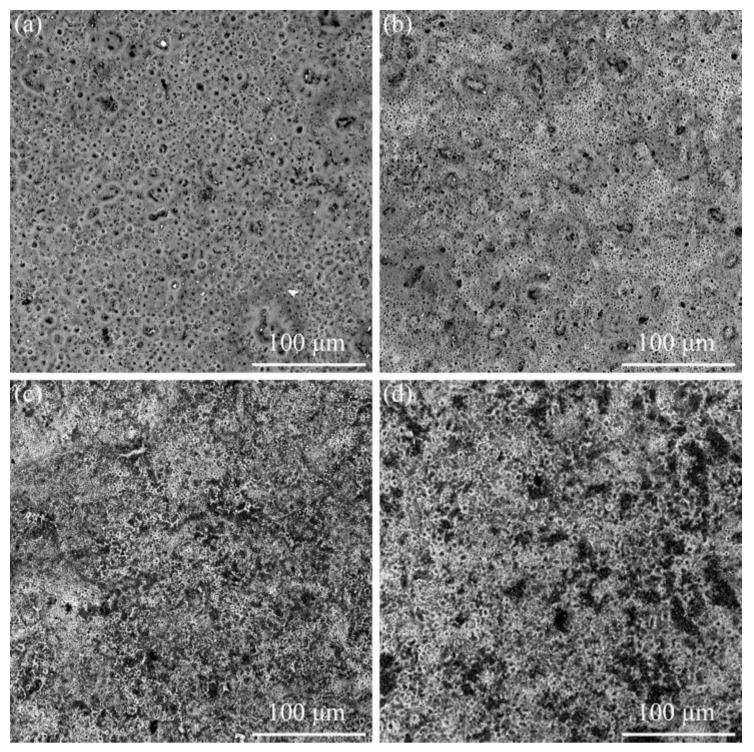
Microstructure of micro-arc oxidation coatings with different α-Al_2_O_3_ concentrations. (**a**) 0 g/L; (**b**) 1 g/L; (**c**) 3 g/L; (**d**) 5 g/L

**Figure 3 nanomaterials-13-01802-f003:**
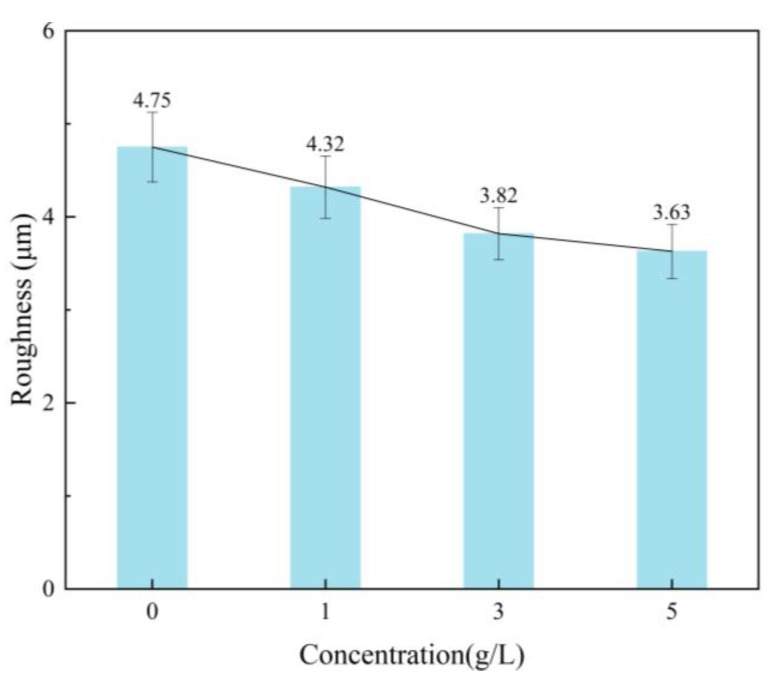
Surface roughness of micro-arc oxidation coatings with different α-Al_2_O_3_ concentrations.

**Figure 4 nanomaterials-13-01802-f004:**
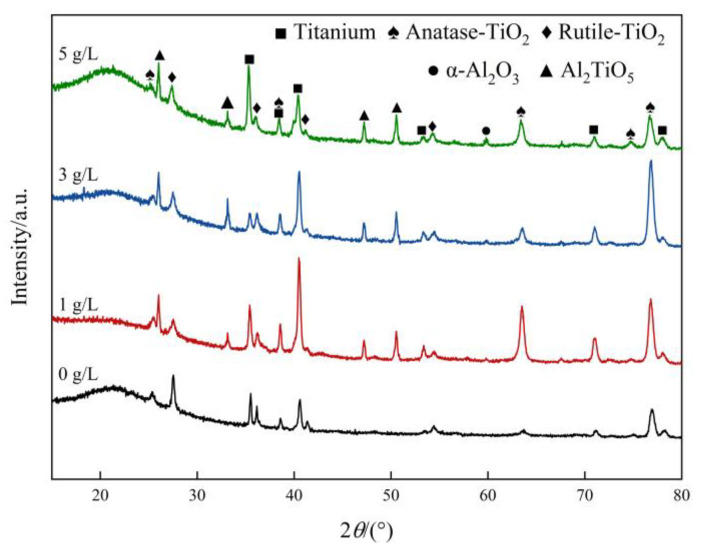
Microstructure of micro-arc oxidation coatings with α-Al_2_O_3_ particle concentrations.

**Figure 5 nanomaterials-13-01802-f005:**
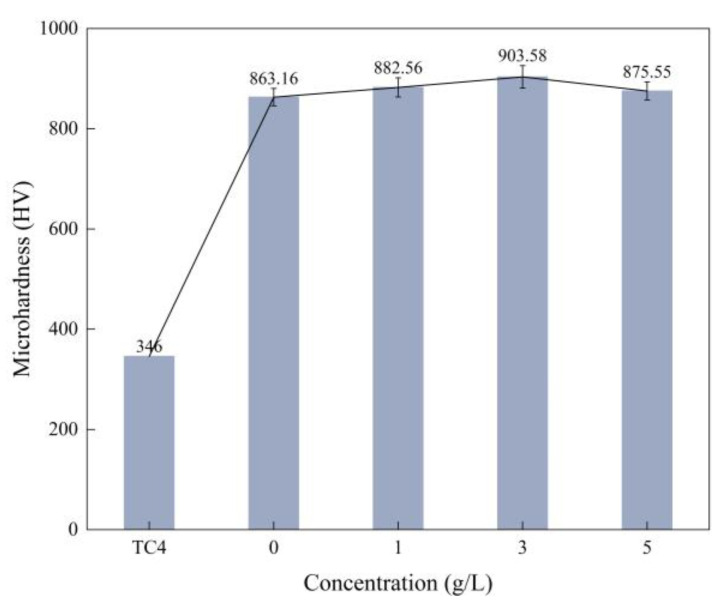
Microhardness of micro-arc oxidation coatings with different α-Al_2_O_3_ concentrations.

**Figure 6 nanomaterials-13-01802-f006:**
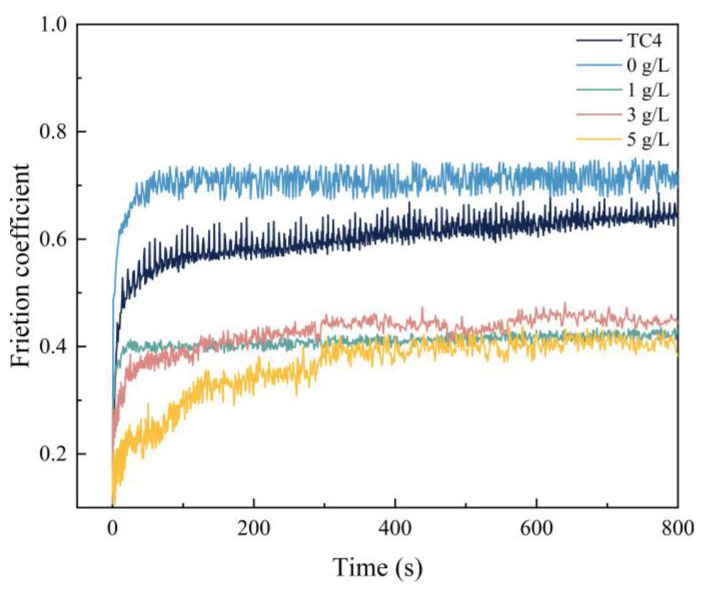
The friction coefficient of micro-arc oxidation coatings with different α-Al_2_O_3_ particle concentrations.

**Figure 7 nanomaterials-13-01802-f007:**
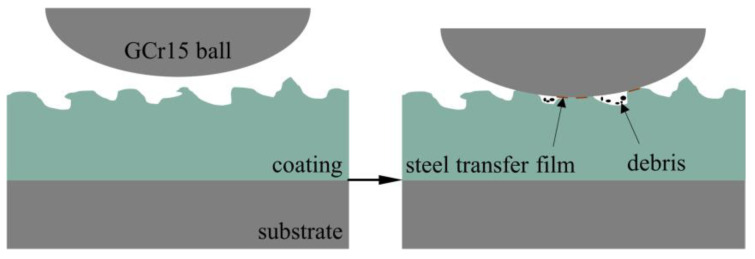
Micro-arc oxidation coating friction mechanism diagram.

**Figure 8 nanomaterials-13-01802-f008:**
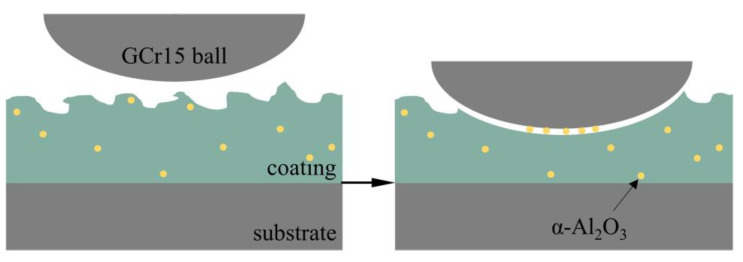
Micro-arc oxidation coating friction mechanism diagram.

**Figure 9 nanomaterials-13-01802-f009:**
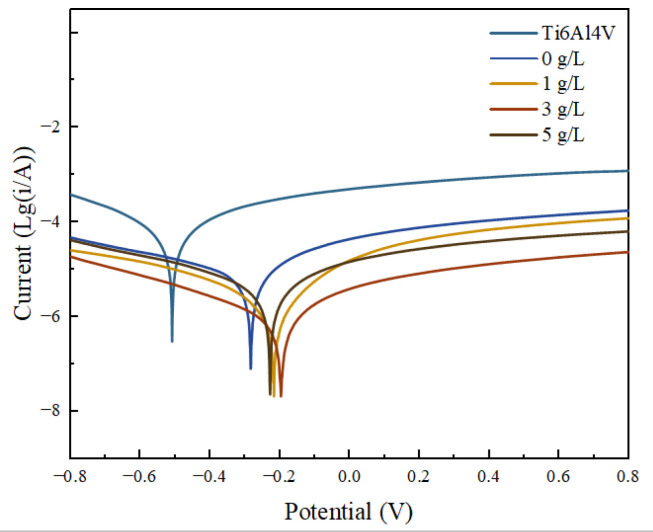
Tafel curves of Ti6Al4V alloy and micro-arc oxidation coating with different α-Al_2_O_3_ concentrations.

**Figure 10 nanomaterials-13-01802-f010:**
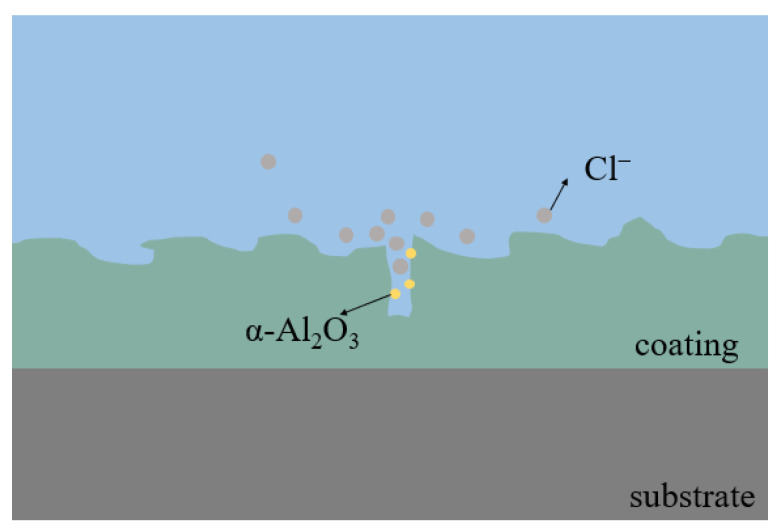
α-Al_2_O_3_ nanoparticles repair the discharge channel.

**Figure 11 nanomaterials-13-01802-f011:**
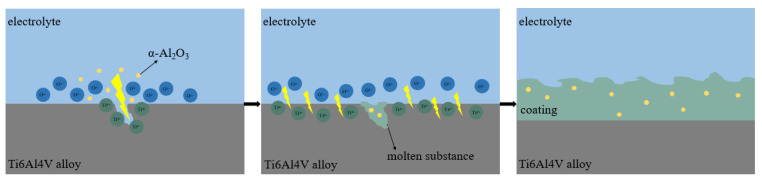
The growth process diagram of micro-arc oxidation coating on Ti6Al4V alloy.

**Figure 12 nanomaterials-13-01802-f012:**
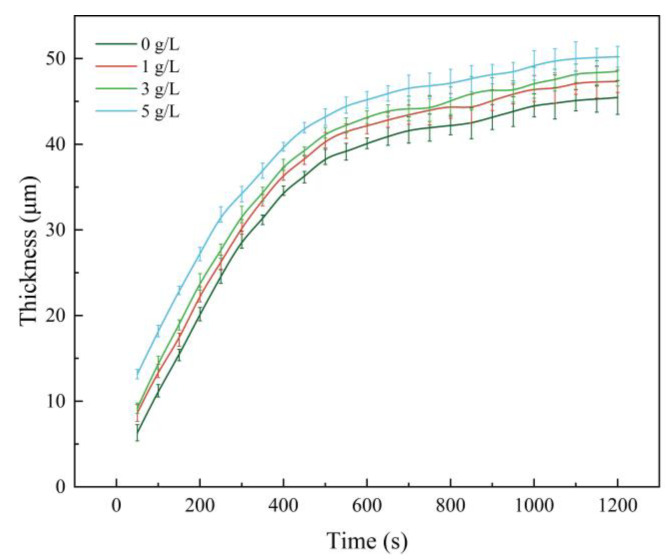
Growth curves of micro-arc oxidation coatings.

**Table 1 nanomaterials-13-01802-t001:** Results of Tafel curve fitting for Ti6Al4V and micro-arc oxidation coatings.

Concentration/(g/L)	*E*_corr_/V	*i*_corr_/(A/cm^2^)	*β*_a_/(V·dec^−1^)	*β*_c_/(V·dec^−1^)	*R*_p_/(Ω/cm^2^)
Ti6Al4V	−0.509	2.25 × 10^−5^	0.20	0.20	1.93 × 10^3^
0	−0.283	6.63 × 10^−6^	0.19	0.31	7.71 × 10^3^
1	−0.217	1.05 × 10^−6^	0.14	0.19	3.30 × 10^4^
3	−0.197	7.99 × 10^−7^	0.19	0.24	5.85 × 10^4^
5	−0.228	2.70 × 10^−6^	0.20	0.23	1.70 × 10^4^

## Data Availability

Data sharing not applicable.

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
