# Peer review of "Effect of α-Al_2_O_3_ Additive on the Surface Micro-Arc Oxidation Coating of Ti6Al4V Alloy"

_nanomaterials, 2023, doi:10.3390/nano13111802_

Round 1
Reviewer 1 Report
The manuscript is devoted to the investigation of the effect of α-Al2O3 on the surface microarc oxidation (MAO) coating of Ti6Al4V alloy. The authors demonstrate that the MAO coating has better wear resistance compared with the Ti6Al4V alloy. The coating prepared with 3 g/L α-Al2O3 particles showed the best frictional wear performance due the lowest roughness and the highest surface hardness. Overall, the manuscript can be recommended for publication in Nanomaterials after a minor revision addressing the following comments.
1. References should be corrected. Please, add page or article numbers, which are currently missing. Also, list all the authors avoiding the use of "et al."
2. Please include typical error ranges for all the values reported in the manuscript. Also, please, reduce the number of digits in the reported values to the actual number of significant digits. At the moment some of the numbers have too many digits exceeding the number of significant digits determined by the experimental error.
English is generally OK, although it would benefit a minor editing related to construction of some sentences.
Author Response
Dear Reviewer, Thank you very much for your suggestion. I have revised the article according to your suggestions.
- In the references, I have listed all the authors' names of the article.
- I have modified some of the data in the text and images in section 3.6 to make them valid digits of experimental error.
Reviewer 2 Report
This publication presents interesting results of the “Effect of α-Al2O3 Additive on the Surface Microarc Oxidation Coating of Ti6Al4V Alloy”. The manuscript was well introduced, and the authors adopted convincing methods with a discussion of the different obtained results.
The English language needs minor improvement. General comments
- Comment 1: The English of this manuscript needs slight improvements.
- Comment 2: The mechanisms underlying the observed variations and correlations should be discussed in the Discussion section.
Other comments
Abstract
Please add the novelty of the paper
M&M
Ti6Al4V - Please use subscript for numbers.
Results& Discussion
Please merge the short paragraphs.
Many statements and explanations are provided without references.
References
Please add more references.
Author Response
Dear Reviewer, Thank you very much for your suggestion. I have revised the article according to your suggestions.
- I have touched up the English expression of the article.
- In the article, I have added the analysis of the changes in the study phenomenon and the response mechanism.
- The innovative points of this article are introduced in the introduction section, and according to your suggestion, I have summarized them and added them to the abstract section.
- Due to space limitations, the conclusion section only presents my main findings. I have revised it according to your suggestion.
Reviewer 3 Report
This is lengthy dull manuscript with many details of low interest to the general reader. It should be abridged twice. The authors should present their most significant result instead of lengthy description experiments.
The reader does not know, what MAO coating is. He need to look to the references. One sentence on MAO coating is requested.
Where did the authors get their reagents including the alloy? No clue.
What is C3H8O3 (line 75) ? The reader has to guess.
The manuscript starts with stating mechanical deficient of Ti alloys, and the authors mainly talk about electrolytes.
The cited literature is presented in unexeptable form.
Check the language, grammar once more.
Author Response
Dear Reviewer, Thank you very much for your suggestion. I have revised the article according to your suggestions.
- Due to the journal length requirement of 4000 words or more, and to make the readers understand the content of the article more clearly, many introductions were made in the experimental section. I have made some deletions according to your suggestions to make the research more focused.
- For the convenience of readers, I have added the introduction of MAO coating.
- The experimental reagents and experimental materials were purchased on the internet at the expense of the supervisor, so they are not mentioned in the article.
- Glycerol, also known as propanetriol, with the chemical formula C3H8O3, was not added to the electrolyte of α-Al2O3, and appeared only as an experimental control group, so it was not described in detail.
- Since the microarc oxidation process is influenced by the composition of the base material, the electrolyte and electrical parameters used vary from material to material. To illustrate the need for this study, the Ti6Al4V alloy, which is widely used in industry, is briefly described, but the focus of the experiments remains on the optimization of the electrolyte for its microarc oxidation process.